## Research Article

chronic cough; dehydration; mucus; hypertonic saline; respiratory droplets

**Author for correspondence:**
*David A. Edwards,
E-mail: dedwards@seas.harvard.edu

# Mouth breathing, dry air, and low water permeation promote inflammation, and activate neural pathways, by osmotic stresses acting on airway lining mucus

David A. Edwards[1]* and Kian Fan Chung[2]

[1]John A. Paulson School of Engineering and Applied Sciences, Harvard University, Cambridge, MA, USA and [2]Experimental Studies Unit, National Heart and Lung Institute, Imperial College London, London, UK

## Abstract

Respiratory disease and breathing abnormalities worsen with dehydration of the upper airways. We find that humidification of inhaled air occurs by evaporation of water over mucus lining the upper airways in such a way as to deliver an osmotic force on mucus, displacing it towards the epithelium. This displacement thins the periciliary layer of water beneath mucus while thickening topical water that is partially condensed from humid air on exhalation. With the rapid mouth breathing of dry air, this condensation layer, not previously reported while common to transpiring hydrogels in nature, can deliver an osmotic compressive force of up to around 100 cm $H_2O$ on underlying cilia, promoting adenosine triphosphate secretion and activating neural pathways. We derive expressions for the evolution of the thickness of the condensation layer, and its impact on cough frequency, inflammatory marker secretion, cilia beat frequency and respiratory droplet generation. We compare our predictions with human clinical data from multiple published sources and highlight the damaging impact of mouth breathing, dry, dirty air and high minute volume on upper airway function. We predict the hypertonic (or hypotonic) saline mass required to reduce (or amplify) dysfunction by restoration (or deterioration) of the structure of ciliated and condensation water layers in the upper airways and compare these predictions with published human clinical data. Preserving water balance in the upper airways appears critical in light of contemporary respiratory health challenges posed by the breathing of dirty and dry air.

## Introduction

The breathing of dry air aggravates the severity of respiratory diseases ranging from asthma and COPD to infections such as influenza and COVID-19 (Ghosh *et al.*, 2015; Mecenas *et al.*, 2020; Romaszko-Wojtowicz *et al.*, 2020; Moriyama *et al.*, 2021). Dehydrated airways promote inflammation (Barbet *et al.*, 1988), mucus production (D'Amato *et al.*, 2018), cilia dysfunction (Ghosh *et al.*, 2015; D'Amato *et al.*, 2018; Moriyama *et al.*, 2021) and triggers of cough (Purokivi *et al.*, 2011; Zanasi and Dal Negro, 2022), notably adenosine triphosphate (ATP) release (Button *et al.*, 2013) and associated acidification of the airways (Zajac *et al.*, 2021). By these phenomena, and the associated reduction of mucociliary clearance of inhaled particles (Ghosh *et al.*, 2015; D'Amato *et al.*, 2018; Mecenas *et al.*, 2020), dry upper airways amplify the health risks of polluted air (Edwards *et al.*, 2021).

Non-communicable respiratory diseases such as asthma and chronic obstructive pulmonary disease are an increasing problem at both ends of the age range in low- and middle-income countries (Troeger *et al.*, 2018). Children are especially vulnerable to respiratory disease, as indicated by asthma attacks in the United States among children relative to the general population (Pate *et al.*, 2021). Among U.S. children, who also tend to suffer from dehydration (Brooks *et al.*, 2017), Black and non-Black Hispanic children are most at risk of asthma, with Black children three to four times more likely to die of asthma than non-Black children (Forno *et al.*, 2009). Children of colour are also most dehydrated among all U.S. children (Brooks *et al.*, 2017).

By their proximity to external air, the upper airways are more prone to dehydration than the lower airways, and dehydrate in many natural ways (Wolkoff, 2018), beyond whole-body dehydration. Mouth breathing (Svensson *et al.*, 2006), heavy breathing as occurs on sustained exercise (Karamaoun *et al.*, 2022) and cold dry inhaled air (D'Amato *et al.*, 2018) all contribute to drying out of the upper airways. Dehydration of the larynx is a special threat, given that it is the site of fastest air flow within the airways on inhalation (Sivasankar and Fisher, 2002). Laryngeal dysfunction associated with cough (Vertigan *et al.*, 2018) has consequently long been associated with the breathing of dry air (Purokivi *et al.*, 2011; D'Amato *et al.*, 2018; Zanasi and Dal Negro,

2022). Increased responsiveness of laryngeal protective reflexes triggered by mechanical or chemical stimuli characterises cough hypersensitivity syndrome (Chung *et al.*, 2022), reflected by an increased sensitivity to tussive stimuli and mediated through a neuropathic process associated with neural inflammation of the vagal sensory neurons in the laryngeal area and in the airway submucosa (Chung *et al.*, 2013). Cough and laryngeal hypersensitivity are often associated with airway conditions such as asthma and rhinosinusitis, and with gastro-oesophageal reflux (Bucca *et al.*, 2011), while also often observed in athletes (Boulet and Turmel, 2019), where chronic dehydration of the airways occurs on account of the high ventilation of exercise, generally through the mouth.

One of the primary roles of the upper airways is to humidify inhaled air (Fronius *et al.*, 2012). Humidification of inhaled air necessitates a steady draw of water to the air/water surface from within the airway lining fluid (ALF), from tissues surrounding the airways and from moisture that is condensed on the ALF during exhalation. Water transport within and around the ALF occurs by the establishment of osmotic pressure differences above and below a mucus layer, and across the airway epithelium between the periciliary layer (PCL), within which cilia protrude from ciliated epithelial cells, and surrounding tissue (Tilley *et al.*, 2015; Bustamante-Marin and Ostrowski, 2017).

The osmotic movement of water through the mucus, a hydrogel (Song *et al.*, 2020), is common to transpiring hydrogels in nature (Etzold *et al.*, 2021), as on the surfaces of plant leaves (Wheeler and Stroock, 2008), where osmotic pressures (Shultze, 2017) can be sufficiently large to pull water against gravity across large vertical distances (Wheeler and Stroock, 2008). The facility of water to transport across hydrogels is characterised by a physicochemical (continuum) transport property of the hydrogel called the *water permeability* (Nishiyama and Yokoyama, 2017), which increases and decreases with various natural phenomena, such as dehydration (which tends to shrink the hydrogel) (Vyazmin *et al.*, 2019). Shrinking and expansion of hydrogels, depending on their ionic nature (anionic or cationic), also occur with acidification (Tomar *et al.*, 2014), and alterations in ion content type and concentration (Tomar *et al.*, 2014).

Condensation layers inevitably exist over transpiring hydrogels (Vyazmin *et al.*, 2019) as can be illustrated by way of a simple example: Imagine a mucus-like hydrogel in contact with air and containing salt ions as well as deposited airborne particles, many of which are small enough to diffuse through the hydrogel pores, and others that are too large. The salt ions, fine and ultrafine particles diffuse with slight to moderate restricted motion, while large particle diffusion is restricted or prevented altogether. Moisture in the air condenses onto the hydrogel surface through natural condensation, as in the sudden sweep of super-saturated air as occurs on an exhalation. This leads to deposition of a mass of water on the surface of the hydrogel, and immediate diffusion of salt ions and small particles from the hydrogel into the condensation layer. An equal and opposite mass of water flows into the membrane. Once enough salt and ultrafine particles have diffused into the condensation layer to eliminate concentration gradients, and an equivalent mass of water has moved by osmosis into the hydrogel, solute concentration in the condensation layer will be the same as in the pores of the hydrogel. On evaporation, the mass of water in the layer diminishes, moving the surface of the condensation layer towards the mucus. Since the salt ions and small particles are partially restricted in their movement into the pores of the hydrogel, their concentration increases above the hydrogel, driving diffusion into the pores, and the reverse movement of water into the condensation

layer, the basic physics of osmosis. To supply water at sufficient rate to meet the rate of evaporation, very large osmotic pressures can arise – osmotic pressure exceeding 5,000 atmospheres have been observed in natural hydrogels during evaporation in 40% RH air (Wheeler and Stroock, 2008). Condensation layers of this kind would appear to play a role in the function (humidification and filtration of inhaled air) and dysfunction (cough, clearance breakdown, vocal dysphonia, among others) of upper ALF, and relate to the rehydration efficacy of inhaled hypertonic salines (see Fig. 1).

We sought a basic, quantitative, continuum understanding of condensation layers as relates to these questions and their implications to respiratory health and the activation of neural pathways (Fig. 1). Previous analyses of water evaporation and heat transfer within human lungs during processes of breathing (see Ferron *et al.*, 1985; Wu *et al.*, 2014; Karamaoun *et al.*, 2018; Haut *et al.*, 2021, and the references therein), while quantifying humidification phenomena of inhaled air and overall evaporative draw from the airways as functions of depth of penetration of dry inhaled air, have omitted consideration of the evaporative physical stresses imparted on the airways by treating ALF as a single homogeneous layer of water with homogeneous transport properties. The existence of the condensation layer, and its relevance to airway function, appears therefore not to have been noted.

We explored airway fluid structure and function in healthy human airways during normal tidal breathing and sustained strenuous exercise [high minute volume ($100 \, l \, min^{-1}$) and duration of approximately 1 h; (Joist *et al.*, 2005)] with a micro-structural model of mucus based on a classical hydrodynamic model presented elsewhere (Anderson and Malone, 1974) (see the Methods section and the Supplementary Material). The results of our research are reported here – along with a reflection on the implication of these results to recent discoveries of the nano- and micro-structural nature of respiratory gas and water exchange.

## Results

### Biophysics model

Evaporation from the airways during normal tidal breathing promotes osmotic movement of water through the ALF towards the air surface by way of an imbalance of osmolytic solutes including deposited particles and salt ions (concentrations of the latter being dominant in airway lining fluid, we refer only to salt ions in our analysis, see Supplementary Material) above and below the mucus layer. We find that this imbalance can be expressed[1] by time-averaged condensation layer and PCL salt ion concentrations relative to the salt ion concentration of surrounding tissue, $C_*$:

$$\overline{C_c} = C_* \left( 1 + \frac{\overline{Q_e \chi}}{PE_{epith}} + \frac{\overline{Q_e \chi}}{PE_m} \right), \tag{1}$$

---

[1]The over bar in Eq. (1) and subsequent equations denotes a time average over sufficiently many identical breaths to establish a pseudo-equilibrium of salt ion distribution across the ALF. Evaporation and ALF transport are characterised by the average mass rate of water evaporation, $Q_e$ ($mg \, s^{-1}$), with $PE_m$ the mass permeability of the mucus ($mg \, s^{-1}$) and $PE_{epith}$ the net (transcellular and paracellular) mass permeability of the epithelium ($mg \, s^{-1}$). See the Supplementary Material for the relationship between the 'mass' and water permeability ($\mu m \, s^{-1}$) as is more conventionally measured.

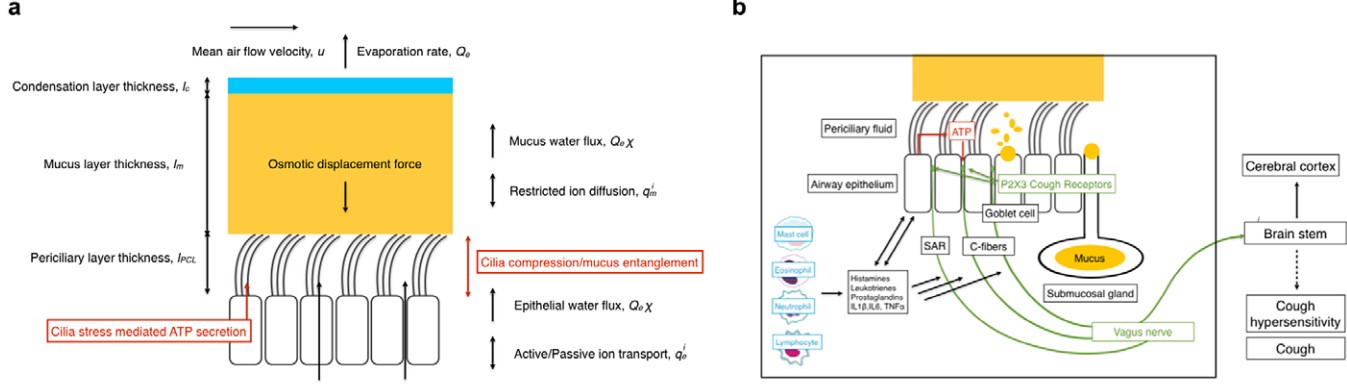

**Fig. 1.** (*a*) Basic one-dimensional airway-lining-fluid geometry, steady-state, time-averaged water fluxes, and osmotic mucus force consequent to the encounter with relatively dry air. The flux of water through the mucus hydrogel ($Q_e \chi$) is proportional to the dehydration factor $\chi$ representing the degree to which the superficial layer of water above the mucus hydrogel that supplies evaporative water on inhalation is not restored on condensation of super-saturated water during exhalation. (*b*) Signalling pathways triggered by dehydration as associated with cilia-compression-activated adenosine triphosphate release.

$$\overline{C_{PCL}} = C_* \left( 1 + \frac{\overline{Q_e \chi}}{PE_{epith}} \right). \qquad (2)$$

The osmotic pressure imbalance ($\Delta \Pi$) created by the difference between Eqs. (1) and (2) pulls water from epithelial cells and surrounding tissues into the PCL and up through the condensation layer in proportion to a condensation factor $\chi$ implicit to which is a dehydration factor $\xi$, as further analysed in footnote 2 and in the Supplementary Material. The osmotic pressure $\Delta \Pi$ displaces the mucus towards the epithelium, reducing the thickness of the PCL over many breaths of inhalation and exhalation time $T$ by an amount

$$\overline{l_{PCL}} = l_{PCL}^0 - \frac{1}{2} \frac{\overline{Q_e} T \chi}{\rho A}, \qquad (3)$$

where $A$ is the cross-sectional area of the airway region in question and $\rho$ the mass density of water. In healthy hydrated airways, the condensation layer thickens to approximately the same extent (see the Supplementary Material for an exact expression), notably (Matsui *et al.*, 2000).

$$\overline{l_C} = l_C^0 + \frac{1}{2} \frac{\overline{Q_e} T \chi}{\rho A} \qquad (4)$$

The mucus layer, therefore, displaces relative to the condensation layer and PCL without loss of ALF volume, whereas, at low values of epithelial or mucus permeability, the condensation layer can thin (see the Supplementary Material) and even recede into the mucus, drying out mucus.

PCL thickness can be restored (or further thinned) by deposition of hypertonic (or hypotonic) saline on the surface of the ALF. Unlike the case of isotonic saline, for which deposition on the surface of ALF increases the condensation layer thickness without increasing PCL thickness and, therefore, is incapable of modulating dysfunction (Marshall *et al.*, 2021), deposition of droplets of mass $M_D$ with hypertonic (or hypotonic) salt ion concentration $C_D$ may

raise (or lower) the tonicity of the ALF sufficiently to hydrate (or dehydrate) the PCL by osmotic water flux from epithelial cells. This mass can be estimated by a simple mass balance. With uniform deposition of a mass of droplets $M_D$ in the upper airways (nose and trachea), the osmotic displacement of the mucus $d_{osm}$ in the direction of the airway lumen can be approximated for salt concentrations up to ~10% tonicity and after many breaths on time average by (see the Supplementary Material)

$$\overline{d_{osm}} \approx \overline{l_{PCL}} \left[ \frac{M_D (C_D - C_*)}{C_* (M_{ALF} + M_D)} \right]. \qquad (5)$$

The evaporative osmotic force ($\Delta \Pi$) pressing down on the PCL reduces cilia motility (Davis and Lazarowski, 2008; Hill *et al.*, 2010) and promotes the secretion of ATP (Davis and Lazarowski, 2008; Button *et al.*, 2013) among other biomarkers (Fig. 1*b*), such as the inflammatory cytokines interleukin-1β, interleukin-6 and tumour necrosis factor-α (Cromwell *et al.*, 1992; Hewitt and Lloyd, 2021), to a degree

$$\overline{C_B} \approx C_B^0 \left( 1 + \alpha_B \frac{\overline{Q_e} T \chi}{A} \right) \qquad (6)$$

that follows from a first-order approximation relative to the base state ($^0$) by a linear relationship between the secreted biomarker ($B$) and the osmotic force or mucus displacement distance (see the Supplementary Material). Here, $\alpha_B$ is a dimensional airway constant ($cm^2 \ mg^{-1}$) that can be directly determined as described below from published data of ATP secretion following the compression of ciliated epithelial cells *in vitro* (Button *et al.*, 2013). Similar first-order approximate relationships can be expressed for diminution of cilia beat frequency (CBF) and elevation of exhaled breath particles (EBPs)[3] (Edwards *et al.*, 1991; Lucassen-Reynders, 1993; Deng *et al.*, 2022) (see the Supplementary Material).

---

[2]As described in the Supplemental Material, the water condensation in the upper airways replenishes approximately 1/3 of evaporated water on inhalation. This fraction is diminished in dehydrated airways by the dehydration factor. The condensation factor therefore represents the fraction of water evaporated from the upper airways that is not supplied by water condensed on exhalation.

---

[3]Loss of ALF volume, as occurs with inadequate permeation of mucus or epithelial layers, increases nonvolatile solute concentration in the ALF as well [see Eq. (1)]. The impact of ALF volume reduction on surfactants present in the ALF is to increase surface elasticity. This enhances the tendency for surface breakup under the shear flow of air that occurs during inhalation (see the Supplementary Material). Scaling in proportion to the Capillary Number ($Ca = \mu u_a \gamma^{-1}$, where $\mu$ is the viscosity of water, $u_a$ a characteristic air velocity and $\gamma$ the surface tension), this phenomenon promotes the generation of respiratory droplets as can be measured in the form of EBPs.

**Table 1.** Airway hydration parameters as a function of breathing parameters and environmental conditions. (1) Airway dehydration factor $\xi$ following the inhalation of dry (10% RH) or moist (60% RH) warm (30 C) air, with an inhalation time $T$ of 1, 2 or 5 s, and a temperature at the carina of 35℃. (2) Water mass evaporated ($Q_eT$) in milligrammes up to the carina (upper airways), and relative humidity at the carina ($RH_{inh}$), on the inhalation of dry (10% RH) or moist (60% RH) warm (30℃) air, with an inhalation time $T$ of 1, 2 or 5 s, and a temperature at the carina of 35℃ (Table 2). (3) Depth of penetration by Weibel airway generation number of unsaturated air, and inhaled air volume (cm$^3$) to saturation beyond the carina ($V_{sat}$), on the inhalation of dry (10% RH) or moist (60% RH) warm (30℃) air, with an inhalation time $T$ of 1, 2 or 5 s, and a temperature at the carina of 35℃. (4) Displacement of mucus or thinning of PCL (micrometres) on the inhalation of dry (10% RH) or moist (60% RH) warm (30℃) air, with an inhalation time $T$ of 1, 2 or 5 s, and a temperature at the carina of 35℃.

| RH | $T = 1$ s | $T = 2$ s | $T = 5$ s | Nose or mouth | Rest or exercise |
|---|---|---|---|---|---|
| $\xi$ (dehydration factor) | | | | | |
| 10.0% | 0.2 | 0.1 | 0 | N | R |
| 60.0% | 0.1 | 0.1 | 0 | N | R |
| 10.0% | 0.2 | 0.2 | 0.1 | M | R |
| 60.0% | 0.1 | 0.1 | 0.1 | M | R |
| 10.0% | 0.3 | – | – | M | E |
| $Q_eT$ (mg), $RH_{inh}$ | | | | | |
| 10.0% | 3.2, 25% | 6.4, 42% | 16, 96% | N | R |
| 60.0% | 1.6, 53% | 3.2, 62% | 8, 89% | N | R |
| 10.0% | 1.2, 14% | 2.4, 20% | 6, 41% | M | R |
| 60.0% | 0.6, 48% | 1.2, 51% | 3, 61% | M | R |
| 10.0% | 12.0, 14% | – | – | M | E |
| Airway generation, $V_{sat}$ (cm$^3$) | | | | | |
| 10.0% | 14, 108 | 13, 100 | 6, 36 | N | R |
| 60.0% | 13, 95 | 13, 90 | 10, 50 | N | R |
| 10.0% | 15, 123 | 14, 108 | 14, 92 | M | R |
| 60.0% | 14, 95 | 14, 92 | 13, 85 | M | R |
| 10.0% | 17, 180 | – | – | M | E |
| PCL thickness change $d$ (μm) | | | | | |
| 10.0% | 0.05 | 0.05 | 0 | N | R |
| 60.0% | 0.02 | 0.04 | 0 | N | R |
| 10.0% | 0.07 | 0.14 | 0.18 | M | R |
| 60.0% | 0.04 | 0.08 | 0.2 | M | R |
| 10.0% | 0.7 | – | – | M | E |

## Condensation layer evolution, ATP secretion and neural activation of asthmatic airways

We sought to determine whether the enhancement of ATP [approximated by Eq. (6)] consequent to the normal breathing of dry air might generate sufficient ATP to elicit cough by activation of P2X3 receptors in hypersensitive (asthmatic) airways (Fowles et al., 2017), and, therefore, whether rehydration of the upper airways [see Eq. (5)] might benefit airway function, and in the case of hypersensitive airways complement the ATP-blocking action of P2X3 antagonist drugs against chronic cough (Abdulqawi et al., 2015; Smith et al., 2020; Zhang et al., 2022). We assumed a base case of the nasal or mouth breathing of warm air (30℃) in circumstances of human breathing ranging from dry (10% RH) to moist (60% RH) air. We assumed in all tidal-breathing cases a tidal volume of 0.5 l with fast ($T = 1$ s), moderate ($T = 2$ s) and slow ($T = 5$ s) breaths. Assuming a 1-s transition from inhalation to exhalation, these cases correspond to a range of 20 breaths per minute ($T = 1$ s) to 5–6 breaths per minute ($T = 5$ s). We also considered the case of mouth breathing during strenuous exercise (increasing the ventilation rate by a factor 10 from the fast-breathing case, i.e., from 10 to 100 l min$^{-1}$). The results of our analysis are summarised in Table 1 (see the Supplementary Material).

Fig. 2 shows the predicted evolution of the condensation layer as a function of the breathing conditions as described in Table 1. Normal (hydrated) permeabilities of the mucus and epithelial cell layers were assumed to be 1,000 μm s$^{-1}$ and 100 μm s$^{-1}$ (see Supplemental Material) (Matsui et al., 2000), recognising airway permeabilities have been reported over a wide range (Schmidt et al., 2017) and that ionic strength, pH, dehydration and mucin secretion can all reduce osmotic membrane permeabilities (Tomar et al., 2014; Etzold et al., 2021). The breathing of dirty air may further diminish mucus water permeability (see the Supplementary Material)

So long as water permeates effectively across epithelial and mucus barriers (normalised water permeability approximately 20% or greater of fully hydrated levels), the consequence of water evaporation from the upper ALF (Fig. 2a) is to slightly thicken the

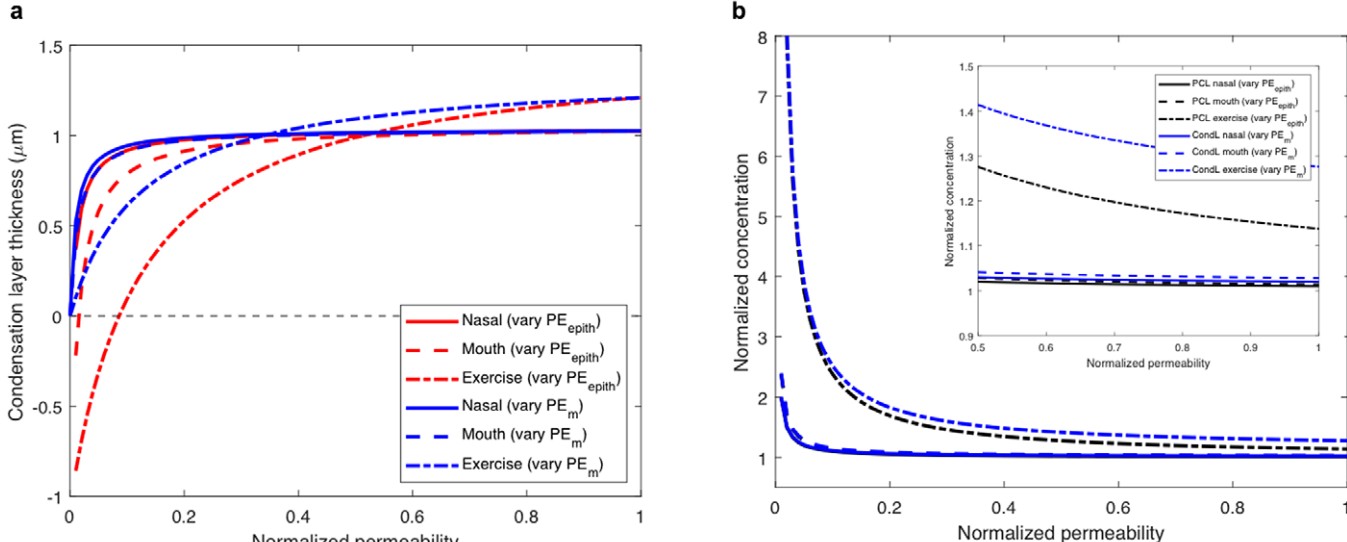

Fig. 2. (*a*) Time-averaged condensation layer thickness as a function of the normalised epithelial (red) and mucus (blue) permeabilities in conditions of fast ($T = 1$ s) breathing of dry (10% RH) air via the nose (solid lines) or mouth (dashed lines) at rest or exercise. (*b*) Time-averaged normalised concentration as a function of the normalised epithelial (Black) and mucus (blue) permeabilities in conditions of fast ($T = 1$ s) breathing of dry (10% RH) air via the nose (solid lines) or mouth (dashed lines) at rest or exercise.

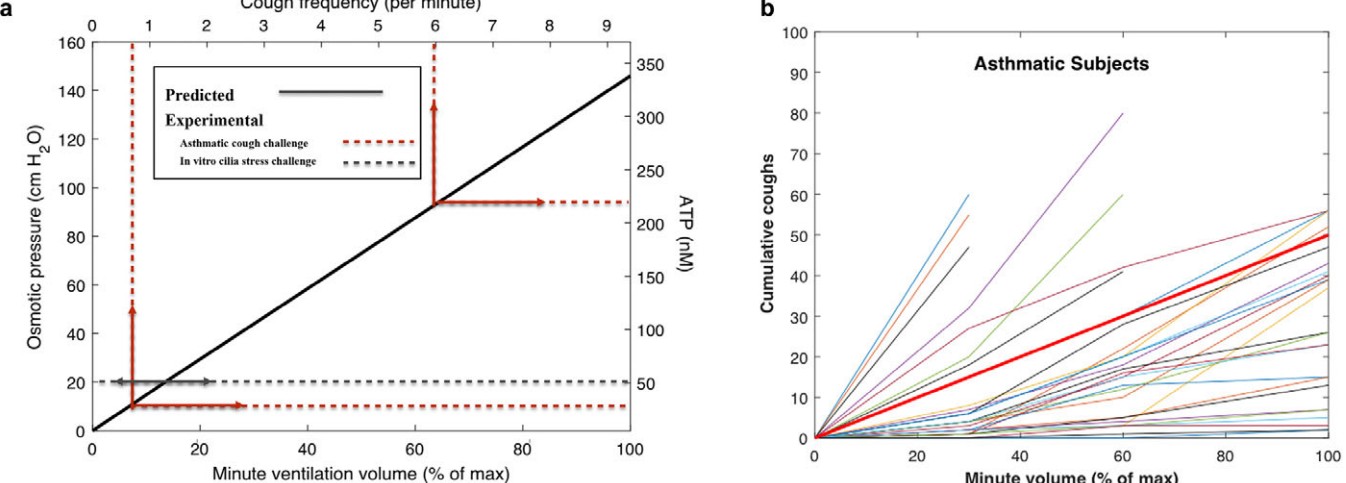

Fig. 3. Predicted and experimentally measured (as reported by Button *et al.*, 2013; Fowles *et al.*, 2017) relationships between minute volume, adenosine triphosphate (ATP) concentration in the airways and cough incidence. (*a*) Predicted (solid Black line) time-averaged steady-state osmotic pressure force acting on the PCL as a function of minute volume (with maximum assumed minute volume of 200 l min⁻¹). The predicted linear relationship between osmotic pressure and ATP concentration is based on the first-order approximate expression Eq. (6) with $\alpha_B = \alpha_{ATP} \sim 245$ cm² mg⁻¹ determined by the measured ATP extracellular concentration (50 nm) and compressive force (20-cm H₂O) reported in Button *et al.* (2013). The predicted linear relationship between (asthmatic airway) CF and ATP concentration is based on the first-order approximate expression Eq. (7) where the unity coefficient is determined by fit of Eq. (7) to the measured CF observed at the two ATP levels shown by the dotted red line by Fowles *et al.* (2017) on topical delivery of ATP at the estimated ALF concentrations depicted in the figure, with the airway lining fluid concentrations being estimated by the delivery efficiencies reported elsewhere (Schlesinger and Lippmann, 1976; Khan *et al.*, 2005). (*b*) Cough incidence (number of coughs) *versus* minute volume as a percentage of maximum minute volume following the deep breathing of dry air. Experimental data are as reported by Purokivi *et al.* (2011) for asthmatic subjects, and the prediction (thick red line) is based on Eq. (7) (Fig. 3*a*) without fitted parameters.

condensation layer (while thinning the PCL to the same degree), thereby conserving overall ALF volume. As breathing rate (minute volume) increases to 100 l min⁻¹ (strenuous exercise), the thickening of the condensation layer reaches around 0.7 μm, which is equivalent to the distance of displacement of the mucus membrane towards the epithelium (Table 1). The salinity of the ALF varies to a small degree (Fig. 2*b*) in all but the high minute-volume case as can be seen by the inset of Fig. 2*b*. During strenuous exercise, the salinity of the condensation layer exceeds that of the PCL, while both salt concentrations are elevated, reaching around 20–40% higher osmolarity than at rest, similar to osmolarity increases that have been measured post

exercise in athletes (Tatsuya *et al.*, 2011). In the case of strenuous exercise, relatively small changes in permeabilities (around 40% of the assumed 'fully hydrated' permeation rate) can lead to loss of condensation layer volume and corollary loss of ALF volume (Fig. 2*a*), and in the case of very low permeabilities, the condensation layer can achieve a negative thickness, meaning that mucus dries out.

The compression that drives PCL thinning [see Eq. (4)] and ATP secretion in the airways [see Eq. (6)] increases linearly with evaporation rate $Q_e$ or ventilation rate. Cough reflex in asthmatics has also been shown to increase in a linear fashion with airway ATP levels (Fowles *et al.*, 2017) following topical deposition of ATP. These

relationships are portrayed with supporting experimental data in Fig. 3a. We determined the linear coefficient characterising ATP concentration increase [see Eq. (6)] as $\alpha_{ATP} \sim 245$ cm$^2$ mg$^{-1}$ by fitting data gathered in direct *in vitro* measurements of ATP secretion levels (50 nm) following compression (20-cm H$_2$O) of cilia beneath a mucus mimetic gel (Button *et al.*, 2013). The linear coefficient characterising the relationship between cough frequency (*CF*) (coughs per minute) and mucus displacement [see Eq. (4)] is determined by matching the linear prediction with human CF data gathered on topical administration of ATP (Fowles *et al.*, 2017), leading to a remarkably simple relationship between CF (coughs per minute) in hypersensitive (asthmatic) human airways and the product of evaporation rate $Q_e$ (mg s$^{-1}$), condensation factor $\chi$ (dimensionless) and inhalation time $T$ (s):

$$\overline{CF_{HS}} \approx \overline{Q_e} T \chi. \tag{7}$$

An identical relationship follows from the data of Fowles *et al.* (2017) for normal (non-asthmatic) airways with CF approximately 10-fold less common. Fig. 3a presents a comprehensive pictorial representation of the biophysical mechanism by which dehydration of the upper airways elicits cough reflex in the hypersensitive airways of asthmatic human subjects.

Fig. 3b compares the predicted CF relationship determined by Fig. 3a with the results of a Finnish study (Purokivi *et al.*, 2011) where 36 asthmatics and 14 healthy human subjects deeply breathed very dry air. Coughs post challenge were monitored in the study for up to 10 min, with primary coughing occurring in the first minute post challenge. Fig. 3b shows the results from this study for the asthmatic group in comparison with the prediction. Our predictions, which involve no fitted parameters, follow from the 30°C, 10% RH case of Table 1, on relatively deep breathing ($T = 2$ s), with the totality of coughs assumed to have occurred within 1 min post challenge. Cough incidence grows linearly as ATP levels grow (Fig. 3a) with increased mechanical stress on the airway epithelium caused by the amplification of airway dehydration accompanying rising ventilation rates. Reasonable agreement is also found (not shown) between the theoretical prediction of CF on the breathing of dry air in normal human subjects (where coughs observed in Purokivi *et al.*, 2011, are approximately an order of magnitude less frequent than with the asthmatics].

### Clearance breakdown and inflammation in healthy airways

Airway dehydration, in otherwise healthy airways, further promotes inflammation, reduction in CBF and elevation of EBP owing to the evolution of the condensation layer portrayed in Fig. 2. Table 2 summarises theory predictions[4] of this airway dysfunction,

as well as the estimated masses of 5% hypertonic saline required to reduce airway dysfunction with a ~10-μm mass median aerodynamic diameter aerosol (Calmet *et al.*, 2019), in the range of breathing circumstances characterised by Table 1.

We compared the predictions of Table 2 with a range of published human clinical data in Fig. 4. Fig. 4a,b shows a comparison with human clinical data gathered in states of strenuous (1 h) exercise as reported in Osaka (Tatsuya *et al.*, 2011), Boston (George *et al.*, 2022) and Munich (Mutsch *et al.*, 2022) studies with predictions based on the case of inhaled air at 30°C and 10% RH.[5] The correlation of prediction and experiment reflects the fitting of these exercise data with the predictions for the determination of the linear perturbation constants.[6] The elevation of inflammatory cytokines in the saliva of the athletes post exercise in the Osaka study (Fig. 4a) parallels the rise in ATP observed with *in vitro* compression of cilia (Button *et al.*, 2013) and as predicted owing to the osmotic stress transmitted to the airway epithelium at high ventilation rate with the mouth breathing of dry air (Fig. 3a). The far greater predicted elevation of ATP (Fig. 3a) with ventilation rate accompanying strenuous exercise (50% of maximum minute volume), relative to the rise of inflammatory cytokines (Fig. 4a), is similar to the rise (approximately eightfold) in EBPs post exercise observed in both the Boston (George *et al.*, 2022) and Munich (Mutsch *et al.*, 2022) studies, suggesting relative EBP rise as an indicator of upper airway dehydration. Fig. 3c,d compares predictions (obtained without fitted coefficients) of CBF and EBP with data obtained in tidal breathing conditions via independent Mannheim (Birk *et al.*, 2017) and Cambridge (Reihill *et al.*, 2021) studies. In the Mannheim study (Birk *et al.*, 2017) (see Fig. 3c), newly tracheostomised patients were treated either with cool (ambient temperature) dry (compressed) air and nebulisation of isotonic saline or with heated (37°C) humidified (100% RH) air 8 h per day for 14 days post tracheotomy. Epithelial tracheal cells were harvested at days 2, 4, 6, 8 and 10 post surgery and CBF measured *in vitro* in both (non-randomised) groups. Nasal CBF diminishes on exercise with dehydration within the range of the predications based on moderate (intermediate to slow and fast) tidal breathing rate. In the Cambridge study (Reihill *et al.*, 2021) (see Fig. 4d), human subjects mouth breathed ambient (25°C) humid (40–50% RH) air for 20 min, then moved into a dry air (10% RH) ambient temperature environment where they remained for 2 h (Fig. 3d). The elevation of EBF on moving from the humid to the dry air environment in the Cambridge (Reihill *et al.*, 2021) study is roughly twofold at tidal breathing, close to the predicted value (see Fig. 3d; see also Table 2).

### Rehydration of the upper airways

Following the breathing of dry air for 2 h, the subjects in the Cambridge study (George *et al.*, 2022) inhaled via the nose approximately 10 mg of 5% hypertonic saline (with either NaCl, or CaCl$_2$ or MgCl$_2$) amounting to an estimated 3 mg deposition in the larynx and trachea based on a mass-median droplet diameter of around 10 μm (Calmet *et al.*, 2019). Rehydration of the upper airways by the nasal inhalation of 5% NaCl, or CaCl$_2$ or MgCl$_2$ reduces EBP to levels reflective of the breathing of humid air (Fig. 4d). The measured reduction of EBP is close to the predicted value based on the 2.5-mg 5% hypertonic saline

---

[4]We determined $\alpha_I \sim 10$ cm$^2$ mg$^{-1}$, $\alpha_{CBF} \sim 11$ cm$^2$ mg$^{-1}$, $\alpha_{EBP} \sim 583$ cm$^2$ mg$^{-1}$, by comparing predictions [see Eq. (6) and Eqs (29), (30) and (32) in the Supplementary Material] of inflammatory cytokine secretion, CBF and EBP with published human clinical data (Müns *et al.*, 1995; Tatsuya *et al.*, 2011; George *et al.*, 2022; Mutsch *et al.*, 2022). We compared predictions in the case of strenuous exercise (Table 2) to the observed increases in salivary cytokines (Tatsuya *et al.*, 2011), the observed decrease in CBF (Müns *et al.*, 1995) and the observed increase in EBP (George *et al.*, 2022; Mutsch *et al.*, 2022) in strenuous exercise studies. The results of these fits are shown in Fig. 4a,b. Our estimates of hypertonic saline deposition (Fig. 4d) are based on the assumption of uniform deposition in the nose, larynx and trachea, and on a 10-μm mass median aerodynamic diameter of the nasally inhaled solution (a 10-μm droplet inhaled via the nose deposits approximately 70% of the inhaled mass in the nose and 30% in the larynx and trachea; see Calmet *et al.*, 2019).

---

[5]See footnote 4.
[6]See footnote 4.

**Table 2.** Airway function versus breathing parameters and environmental conditions. (1) Airway adenosine triphosphate (ATP) concentration following the inhalation of dry (10% RH) or moist (60% RH) warm (30°C) air, with an inhalation time $T$ of 1, 2 or 5 s, and a temperature at the carina of 35°C. (2) Airway inflammatory cytokine concentration following the inhalation of dry (10% RH) or moist (60% RH) warm (30°C) air, with an inhalation time $T$ of 1, 2 or 5 s, and a temperature at the carina of 35°C. (3) Cilia beat frequency diminution $(1 - CBF)/\alpha_{CBF}$ following the inhalation of dry (10% RH) or moist (60% RH) warm (30°C) air, with an inhalation time $T$ of 1, 2 or 5 s, and a temperature at the carina of 35°C. (4) Breakup factor $B$ increase $(1 - B)/\alpha_B$ on the inhalation of dry (10% RH) or moist (60% RH) warm (30°C) air, with an inhalation time $T$ of 1, 2 or 5 s, and a temperature at the carina of 35°C. (5) Mass of hypertonic saline $(C_D/C_* = 5)$ needed to restore the PCL thickness following the inhalation of dry (10% RH) or moist (60% RH) warm (30°C) air, with an inhalation time $T$ of 1, 2 or 5 s, and a temperature at the carina of 35°C. (5) Mass of hypertonic saline $(C_D/C_* = 5)$ needed to restore the PCL thickness following the inhalation of dry (10% RH) or moist (60% RH) warm (30°C) air, with an inhalation time $T$ of 1, 2 or 5 s, and a temperature at the carina of 35°C.

| RH | T = 1s | T = 2s | T = 5s | Nose or Mouth | Rest or Exercise |
|---|---|---|---|---|---|
| $(C_{ATP}-C_{ATP}^{0})/C_{ATP}^{0}$ | | | | | |
| 10% | 1.9 | 2.0 | 0 | N | R |
| 60% | 0.1 | 0.5 | 0 | N | R |
| 10% | 2.0 | 3.0 | 6.0 | M | R |
| 60% | 0.5 | 0.5 | 1.0 | M | R |
| 10% | 9.0 | — | — | M | E |
| $(C_I-C_I^{0})/C_I^{0}$ | | | | | |
| 10% | 0.2 | 0.4 | 0 | N | R |
| 60% | 0.02 | 0.1 | 0 | N | R |
| 10% | 0.4 | 0.6 | 1.2 | M | R |
| 60% | 0.1 | 0.1 | 0.2 | M | R |
| 10% | 1.8 | — | — | M | E |
| $(CBF_0-CBF)/CBF_0$ | | | | | |
| 10% | 0.1 | 0.2 | 0 | N | R |
| 60% | 0.01 | 0.05 | 0 | N | R |
| 10% | 0.2 | 0.3 | 0.7 | M | R |
| 60% | 0.07 | 0.07 | 0.2 | M | R |
| 10% | 1.0 | — | — | M | E |
| $(EBP-EBP_0)/EBP_0$ | | | | | |
| 10% | 1.6 | 3.1 | 0 | N | R |
| 60% | 0.7 | 1.3 | 0 | N | R |
| 10% | 3.1 | 3.8 | 11.6 | M | R |
| 60% | 1.3 | 1.3 | 2.7 | M | R |
| 10% | 15.5 | — | — | M | E |
| $M_D$ (mg) | | | | | |
| 10% | 0.3 | 0.3 | 0 | N | R |
| 60% | 0.2 | 0.2 | 0 | N | R |
| 10% | 0.5 | 0.3 | 0.3 | M | R |
| 60% | 0.3 | 0.2 | 0.2 | M | R |
| 10% | 5.0 | — | — | M | E |

case of Table 2 relative to the dry air (10% RH) case reflecting the impact of rehydrating the PCL via the osmotic action of hypertonic salts on the epithelial cell layer. While the three salts show similar degrees of suppression (George *et al.*, 2022) as predicted by the osmotic rehydration mechanism (Fig. 2), clearance times, and therefore duration of rehydration, will vary between the salts, the monovalent sodium clearing more rapidly than the divalent calcium and magnesium, as reported in the Cambridge study (George *et al.*, 2022) and has been reported elsewhere (George *et al.*, 2020; Field *et al.*, 2021).

## Discussion

We find that the osmotic movement of water necessary to support humidification of relatively dry inhaled air produces a net physical force originating in a condensation layer that presses mucus towards the airway epithelium, stressing cilia, secreting ATP and activating neural pathways (Figs 1 and 3). This stress can engender cough in situations of cough hypersensitivity (Fig. 3*b*), and inflammation and clearance breakdown in relatively healthy airways (Fig. 4).

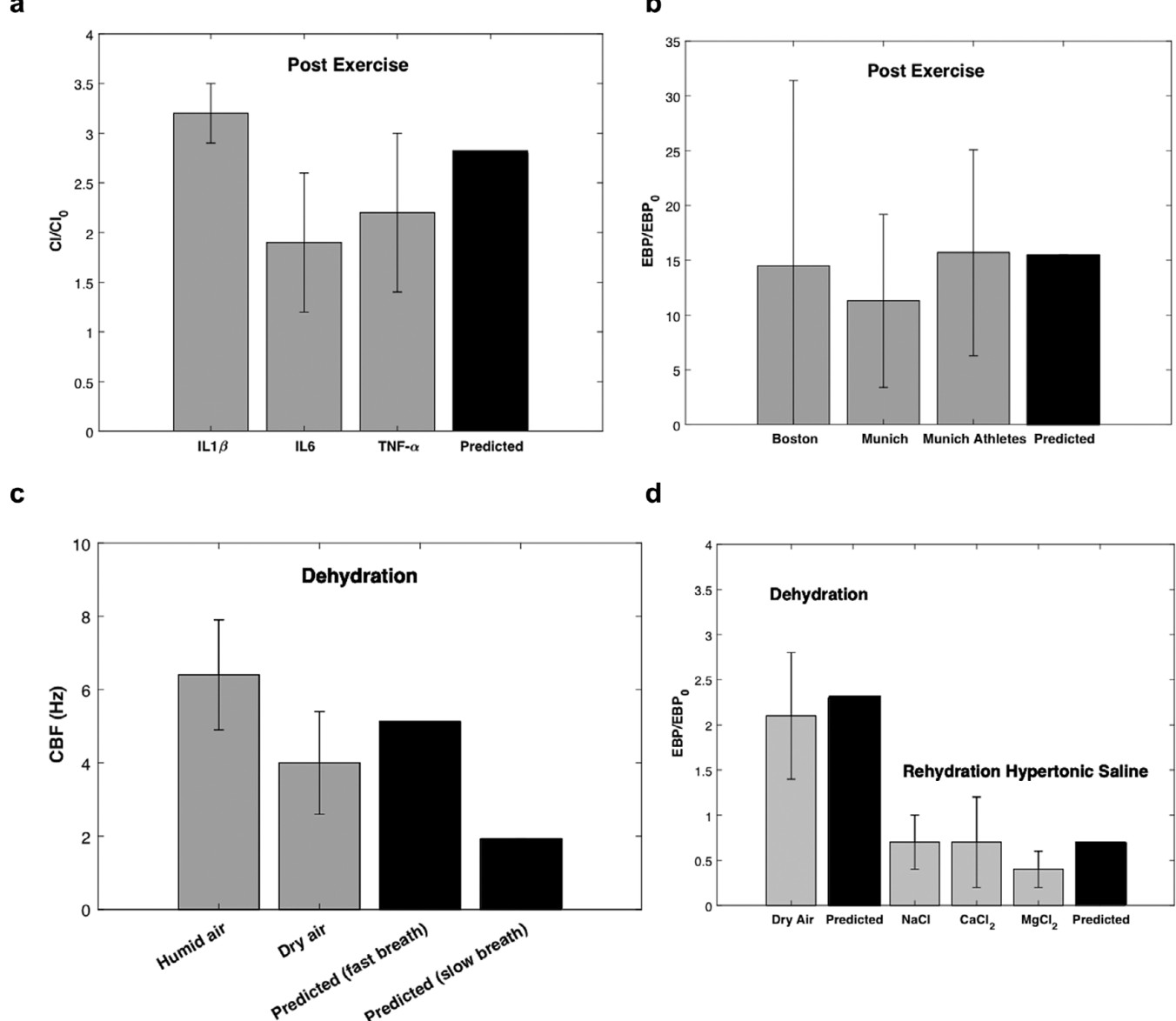

**Fig. 4.** (*a*) Post-exercise exhaled inflammatory marker concentration (CI) relative to pre-exercise. Data represent mean values (grey boxes) with standard deviations reported from Osaka study (Tatsuya *et al.*, 2011). Prediction (Black box) is based on Eq. (36) of the Supplementary Material in the case of exercise (Table 2). (*b*). Post-exercise exhaled breath particles (EBPs) relative to pre-exercise. Data represent mean values (grey boxes) and standard deviations as reported from the Boston study (George *et al.*, 2022), the Munich study (all participants) (Mutsch *et al.*, 2022) and the Munich study (seasoned athletes only) (Mutsch *et al.*, 2022). Prediction (Black box) is based on Eq. (38) of the Supplementary Material in the case of exercise (Table 2). (*c*) Cilia beat frequency post exposure to dry air (with isotonic saline) or perfectly humid air in patients following tracheotomy as reported by Birk *et al.* (2017). Prediction (Black box) and standard deviation is based on the dry air (10% RH) mouth-breathing case of Table 2 with maximum of 20 and a minimum of 6 breaths per minute (i.e. 1 or 5 s inhalation times). (*d*) EBP post exposure to dry air (10% RH) relative to EBP in humid room conditions (40–50% RH) (Dehydration) (grey box) and following the delivery of 5% hypertonic saline to the upper airways (Rehydration) (grey boxes). Data represent mean values and standard deviation as reported from the Cambridge study (Reihill *et al.*, 2021). Prediction is based on Eq. (38) of the Supplementary Material in the case of fully hydrated airway lining fluid [equivalent to the case of slow ($T = 5$ s) nose breathing].

Previous theoretical considerations of the humidification of inhaled air have focused on predicting degrees of humidification of inhaled air as a function of ambient temperature and humidity in a wide range of breathing conditions (see Ferron *et al.*, 1985; Wu *et al.*, 2014; Karamaoun *et al.*, 2018; Haut *et al.*, 2021, and the references therein). By assuming airway fluid to be a homogeneous medium without osmotic restriction of ion and particle transport, previous analyses have omitted consideration of the physical stresses imparted on the airways by water evaporation. While the condensation of moisture on the upper airways during exhalation

has been noted and analysed [see e.g. Haut *et al.*, 2021], the accumulation of the condensed water on top of mucus owing to the immediate diffusion of osmolytes from the mucus into the condensation layer, and the restricted movement of these osmolytes back into the 'transpiring hydrogel' of mucus, has been ignored, if only because the role of mucus in the regulation of water transport has not previously been considered. This has left unaddressed the question as to how, or if, physical stresses imparted to the airways by natural water evaporation are significant, and, therefore, whether these stresses might underpin phenomena of airway

dysfunction that have long been reported as associated with the breathing of dry air (Barbet *et al.*, 1988; Purokivi *et al.*, 2011; Ghosh *et al.*, 2015; D'Amato *et al.*, 2018; Mecenas *et al.*, 2020; Romaszko-Wojtowicz *et al.*, 2020; Moriyama *et al.*, 2021). Our conclusion that these phenomena are indeed significant enough to alter ALF structure and function is consistent with general physical understanding of 'transpiring hydrogels' in nature (Wheeler and Stroock, 2008; Etzold *et al.*, 2021).

The 'condensation layer' protects mucus from drying out by growing in thickness as the mucus displaces towards the epithelium (Fig. 2). In circumstances of poor epithelial and/or mucus water permeation, the condensation layer can recede into the mucus and disappear as topical mucus moisture, a condition we find occurs most readily on the mouth breathing of dry air at high minute volume – combining three of the frequent contemporary conditions that dehydrate the airways, an extreme case of which is strenuous exercise on a cold day (Fig. 2). The circumstance of poor water permeation arises in the disease condition of cystic fibrosis, where defects in the CFTR gene prevent chloride transport across the apical membrane of epithelial cells (Weiser *et al.*, 2011), and between cells via the paracellular route (Weiser *et al.*, 2011), thereby preventing water transport into the airways. CF has long been studied as a condition of chronic and excessive airway dehydration (Boucher, 2007). Among therapeutic approaches to relieve the symptoms of cystic fibrosis, rehydration of the airways by the delivery of hypertonic (NaCl) saline (Kelly *et al.*, 2021) has proven particularly useful, if of shorter duration than is observed with hypertonic salines of divalent salts ($CaCl_2$ and $MgCl_2$) (Edwards *et al.*, 2004; George *et al.*, 2020; Field *et al.*, 2021), which nevertheless benefit rehydration of the upper airways to similar degrees (Fig. 4*d*), given that all the salts act by the same physical airway hydration mechanism.

Our analysis indicates that the topical delivery of hypertonic saline to the upper airways will hydrate the PCL after many breaths and in the steady-state conditions of the analysis, thereby reducing pressure on underlying cilia and reducing ATP secretion [see Eq. (5), Table 2 and Fig. 4*d*]. It appears, therefore, that the delivery of hypertonic salines to the upper airways with a droplet size of approximately 10 µm (i.e. in the approximate range of 8–15 µm) may provide an effective means of hydrating the upper airways and reducing propensity to cough in those with hypersensitive cough syndrome. By reducing the ATP trigger of P2X3 cough receptors, upper airway hydration with hypertonic salines, and particularly with divalent (calcium and magnesium) salines that retain hydration effects longer than monovalent (sodium) salines (Edwards *et al.*, 2004; George *et al.*, 2020; Field *et al.*, 2021), might also be a helpful adjuvant to the efficacy of P2X3 antagonist drugs. The combination of benefits of upper airway hydration with hypertonic salts implied by the relationship between airway dysfunction and upper airway hydration state indicated in Figs 3*a* and 4*a–d* suggests that the hydrating of the upper airways may confer in normally hydrating airways at risk of respiratory illness prophylactic and treatment benefits (George *et al.*, 2022) akin to the benefits conferred in CF airways by full-lung airway hydration with hypertonic salines (Kelly *et al.*, 2021).

Consistent with our observations is that topical administration of *hypotonic* saline has previously been shown to *induce* cough (Ventresca *et al.*, 1990). Notably, in (Ventresca *et al.*, 1990), pure water was administered by a nebuliser for approximately 1 min to non-asthmatic human subjects, with an output (around 20 mg per breath) suggesting ±60 mg of saline deposited in the trachea. This is approximately the quantity of water in tracheal ALF, suggesting

from Eq. (6) a loss of approximately 50% of PCL thickness. This dramatic thinning of the PCL by the prolonged nebulisation of water will produce approximately five times the stress on cilia relative to the case analysed here of strenuous exercise and the breathing of dry air (Table 2) – and is, therefore, consistent with the observation that even normal human subjects tend to cough significantly on the breathing of pure water droplets or hypotonic aerosols.

There are many limitations to our analysis. We have used a continuum mechanics analysis to study water exchange between ALF and inhaled/exhaled air. Osmotic forces are themselves necessarily continuum principles, reflecting the restricted movement of solutes (ions, particles and other osmolytes) through a semi-permeable membrane (the mucus and airway epithelium in our model) (Anderson and Malone, 1974). The analysis does not, therefore, explicitly address the substructure of matter, leaving unaddressed the molecular composition of the condensation layer in the upper airways. Recent discoveries (Ninham *et al.*, 2022) of the role played by pulmonary surfactant in the movement of gases between the alveolar lumen and the systemic circulation suggest that the absorption of oxygen and nitrogen from inhaled air may significantly occur by way of nano-bubbles of oxygen and nitrogen formed in a lattice of pulmonary surfactant, on inhalation, seemingly transitioning to carbon dioxide and water on exhalation (Ninham *et al.*, 2022). Such nano-bubbles have an even better documented role in the endothelial surface layer (Reines and Ninham, 2019), where they reside within the glycocalyx (Reines and Ninham, 2019) and facilitate the oxygenation of blood. Nano-bubbles, whose dimensions range between 4 nm to approximately 40 nm, appear to have extremely small surface tension, permitting the containment of gas at realistic pressures. Conceivably, nano-bubble breakage on the expansion of the lattice structure of pulmonary surfactant in the small airways may contribute to small-airway respiratory droplet formation whose origin has long been held to occur to the 'necking' of surfactant between small airway walls (Scheuch, 2020). The water that condenses on the upper airways on exhalation may then contain molecular and structural elements of nano-bubble forms in the alveolar region of the lungs and promote another 'salt effect' on the composition of the condensation layer that we do not consider in our study. That is, nano-bubbles are stabilised at a critical salt concentration of approximately 0.17 M (Zhou *et al.*, 2021; Ninham *et al.*, 2022), slightly above isotonic salt concentration, while this number is suppressed with convection (Nguyen *et al.*, 2012), as occurs along the surface of ALF in the upper airways. Should nano-bubbles exist in the condensation layer, the elevation of salt concentration that occurs on dehydration (Fig. 2b) might trigger their stabilisation and contribute to the lowering of surface tension and the frothiness of upper airway condensation layer that is observed to occur when the upper airways are dry (Fig. 4) (see Edwards *et al.*, 2004; George *et al.*, 2020; Field *et al.*, 2021). While our continuum analysis has permitted resolution of osmotic forces and prediction of macroscopic displacement of mucus, stresses on underlying epithelia and dysfunction as in the promotion of cough reflex, to understand macroscopic coefficients such as appear in Eq. (6) (and related relationships for CBF and EBP), molecular and cellular substructure and dynamics will need to be considered, and it is intriguing that the evolving understanding of gas and water exchange in the small airways may lead to new insights in the structure and function of the upper airways as well.

Our estimates of airway dysfunction are based on very small perturbations in PCL thickness and are therefore insufficient for estimating dysfunction in severely dehydrated airways. Our assumption of warm equatorial air simplified energy considerations, and avoided an obvious circumstance where further airway dehydration occurs, notably on the breathing of cold air (Barbet *et al.*, 1988; D'Amato *et al.*, 2018; Mecenas *et al.*, 2020). Breathing cold air leads to air of very low water content (similar to our 10% RH case) while also requiring heating of inhaled air in the upper airways, which reduces evaporation loss, and has been well studied in previous work [see e.g. Haut *et al.*, 2021]. In our analysis, we assumed effective ion exchange between compartments of the ALF and therefore ignored a special circumstance as occurs in cystic fibrosis where the inability for chloride ions to transport across the epithelial membrane blocks water movement into the airways. This case of very low 'effective' epithelial water permeability results in circumstances that in practice resemble closely what is revealed in Fig. 2 in the circumstance of vanishingly small epithelial water permeability. The mucus dries out, and eventually the PCL does as well, resulting in a thickening of the mucus and an adherence of the mucus against the apical epithelial wall. We did not consider the shrinkage that can occur of the mucus layer owing to the dehydration of mucus, and the evolution of ionic content of the mucus, as can arise in relatively dehydrating breathing circumstances. This phenomenon has recently been studied and shown to produce shrinkage up to approximately 40% of a characteristic gel diameter, meaning a volume reduction of an order of magnitude or more. While our elucidation of condensation layer biophysics aligns with many general observations of airway hydration, dehydration and dysfunction, further comparison of model predictions with experiments is needed, and assumptions inherent in the model should be relaxed and consequences explored.

Humidifying inhaled air in conditions other than the slow nasal breathing of relatively warm and moist air can harm respiratory health by upper airway structural and function challenges aggravated by the breathing of dirty air. We find that these challenges – originating in osmotic stresses exerted by a condensation layer of water that wets airway lining mucus – can be offset by a combination of effective breathing habits, the humidification of inhaled air and the rehydration of the upper airways by hypertonic salt aerosols. Maintaining proper hydration of the upper airways may be necessary to combat the global respiratory health crisis associated with the breathing of contaminated air.

## Methods

Our analysis of the biophysics of the condensation layer, and of the physical structure of ALF, during processes of breathing is based on the fundamental assumption that solutes within the ALF transport relatively rapidly across transport barriers of the epithelium and mucus on the time scale of the many breaths required to establish a steady-state time-averaged structure in a particular breathing condition. This assumption implies healthy relatively hydrated airways wherein epithelial transporters properly function and airway mucus is not particularly dehydrated. We further assume sufficiently small differences in temperature between the outside environment and body temperature such that heat transfer occurring on the evaporation and condensation of water in the airways has a small impact on the overall rates of water evaporation and condensation. These and other assumptions are listed in Table 3, and the full analysis is provided in the Supplementary Material.

We assumed perfect mixing of air and water within the 'compartments' of the nose (and nasal pharynx), trachea (and larynx) and bronchi (and bronchioles) and used upper airways dimensions as reported in the literature for human airways, whereas from the carina to the small airways, we assumed Weibel lung geometry (see the Supplementary Material). We assume warm ambient air at 30°C. We assumed the gradient of temperature from the outside air to the carina to be such that outside ambient temperature exists up to the carina at which point the air temperature rises to 35°C. This gradient is approximately as has been observed in thermal mapping studies (Nguyen *et al.*, 2012) on the breathing of room temperature (25°C) air at variable ventilation rates, notably where the nose and trachea temperatures ranged from 29 to 32°C and the main bronchi temperatures from approximately 34 to 36°C. All of our mass transport considerations assumed steady or nearly steady-state flow conditions representing time averages of inhalation, exhalation and many cycles of repetitive tidal breathing. We assumed the base case of warm air (30°C) with high (60%) and low (10%) relative humidity (RH), and inhalation and exhalation times of 1, 2 or 5 s, ranging from fast to slow breathing. We also considered a case of 'exercise', as a special case of fast mouth breathing with a velocity of air flow 10 times elevated relative to normal breathing.

Given the very thin ALF relative to the curvature radii of the airways (ranging from hundreds of microns to several millimetres), we assumed the ALF to be essentially flat and of 'infinite' lateral dimension, with the principal physics of water and ion transport occurring in one dimension (Fig. 1). Fully hydrated ALF is assumed to be 30-micron thick in the nose and tracheal compartments, and 10-micron thick from the carina to the small airways, with a PCL of structured water of approximately 7-micron thickness, covered by a mucus hydrogel layer whose structure is static, over which is a thin layer of water from which water is directly exchanged with inhaled and exhaled air. This condensation layer of water will inevitably be thickest following the slow exhalation of relatively warm humid air and thinnest following the rapid inhalation of relatively cool dry air. We assumed a steady-state fully hydrated condensation layer thickness of approximately 1 μm given the water content of supersaturated state of exhaled air during an exhalation into a relatively warm trachea.

As principal air flow in the nasal cavity (approximately 12-cm length from the tip of the nose to the nasal pharynx) occurs in the narrow air passage of the middle or inferior meatus (approximately 0.2 cm radius, or approximately 10-cm$^2$ surface area), we assume relatively quiescent conditions over the majority of ALF surface area within the nose (approximately 160 cm$^2$) over the course of inhalation on normal tidal breathing. Principal air velocity in the trachea (approximately 12-cm length from the larynx to the carina) being driven by the jet of air that emerges from the larynx (with typical peak air velocity on inhalation of approximately 3 m s$^{-1}$), we assume an average velocity on inhalation in the trachea of around 1 m s$^{-1}$. Finally, we assumed salt concentrations in hydrated ALF are approximately equivalent to blood concentrations, such that in the reference state ($C_0$), osmolarity equilibrium has been obtained between the airways and surrounding epithelial cells and vascularised tissues. Table 3 summarises the approximate total masses of principal salt ions and water in the nose and trachea in a fully hydrated state.

The pore-level model of transport in the mucus hydrogel summarised in the Supplementary Material uses a periodic porous medium hydrodynamic model of osmosis as developed elsewhere

**Table 3.** Base assumptions of biophysical model of hydration and dehydration of the human airways. See Supplementary Table S1 for a summary of the Weibel model characteristics on which all calculations in the central airways are based.

| Assumptions | Values | Description of environmental, mass and energy transport, and basic physical constant assumptions |
|---|---|---|
| **Environmental** | | |
| Temperature | 30°C | Representative of warm equatorial air. |
| Relative humidity | 10–60% | Typical moderate range of low to high humidity. |
| **Mass transport** | | |
| Three compartments | | Nose, trachea and central airways are perfect mixing compartments. |
| Steady state | | All transport happens at or near a steady state in real time or over time average. |
| Perfect mixing | | Airborne water mass in each compartment is perfectly mixed. |
| Ion permeation | | Salt ions permeate the mucus membrane with slightly restricted movement owing to mucin surface interactions. |
| Quiescent dominant | | Water evaporation over the entire surface area of the nose is dominated by quiescent evaporative losses. |
| Convective dominant | | Water evaporation over the entire surface area of the trachea is dominated by convective evaporative losses. |
| Trachea air velocity | 1 m s$^{-1}$ | This value assumes that air flow over the surface of the trachea is dominated by the laryngeal jet of air (peak 3 m s$^{-1}$). |
| **Energy transport** | | |
| Upper airways | | The environmental air temperature (30°C) is retained up to the end of the trachea at the carina. |
| Central airways | | From the carina to the last generation prior to full saturation of the air, a temperature of 35°C holds. |
| Evaporation energy | | The loss/gain of energy from the airway lining fluid (ALF) on evaporation/condensation has minor impact on mass water flows. |
| **Physical UA** | | |
| Tracheal | $R = 1$ cm, $L = 12$ cm | Characteristic of typical reported human tracheas. |
| Upper airway areas | 160 cm$^2$ (nose), 60 cm$^2$ (trachea) | Characteristic of typical reported human nose and tracheal dimensions. |
| ALF thicknesses | 30 μm (total), 7 μm (PCL), 23 μm (mucus) | Within the range of reported ALF thicknesses in hydrated human upper airways. |
| ALF volumes | 0.48 cm$^3$ (nose), 0.18 cm$^3$ (trachea) | |
| ALF salt cation masses | 5.5 mg Na$^+$, 70 μg Ca$^{++}$, 7 μg Mg$^{++}$ | Based on the assumption of equilibrium of ALF salts with well-hydrated (isotonic salt) concentration in surrounding tissue. |

(Anderson and Malone, 1974). Mucus is modelled as a porous medium with infinitely long cylindrical pores (i.e. the radius of the pores, $R$, is much smaller than the length of the pores or the mucus thickness, $L$). Each pore is identical to the other, and of an effective diameter deduced from the model by comparison of predictions with reports of experimentally measured ALF permeability values. The model shared in the Supplementary Material explicitly considers transport of salt ions together with deposited airborne particles. The model demonstrates that while restricted motion is far greater for nanoparticle than for salt diffusion within mucus, the high concentration of salts renders the net osmotic pressure by salt ions much greater than that of inhaled particles at least in conditions where cilia continue to beat. We speculate on the possibility of diminished water permeability within mucus owing to the breathing of dirty air and the roles this diminished permeation might have in the phenomena described in Figures 3 and 4. We deduced the dimensions of the model micro-structure of

mucus by comparing the formulas derived in the Supplementary Material for permeability, reflection coefficient and other transport characteristics.

We determined the values of the model parameters in Table 1 by using the data summarised in Table 3 (see also Table S1 in the Supplementary Material for a summary of central airway Weibel geometry), and we determined the unknown variable needed for the determination of the dysfunction parameters appearing Table 2 by comparison with experimental data.[7] The perturbation assumptions by which we determined Eq. (6) and other airway inflammation measures are detailed in the Supplementary Material.

**Open peer review.** To view the open peer review materials for this article, please visit http://doi.org/10.1017/qrd.2023.1.

---

[7]See footnote 4.

**Supplementary materials.** To view supplementary material for this article, please visit http://doi.org/10.1017/qrd.2023.1.

**Conflict of interest.** D.A.E. is a cofounder and shareholder of Sensory Cloud Inc. and a cofounder (while not a shareholder) of Pulmatrix. Each of these companies advances science, technology, and products for consumer health or therapeutic use by the inhalation of aerosols.

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
