## [Reviewer Report]

*Comments to Author*: I have examined with great interest the manuscript, “Breathing dry air promotes inflammation, activates neural pathways and disrupts clearance by osmotic stresses imparted on airway epithelia” because it claims to explain the rise of respiratory diseases which the authors attribute to climate change. This is attributed to chronic dehydration of the airways. To me, this is a bold claim. I do not find it supported by the evidence presented.

So, let’s turn to what is evidence is presented. It is found that water evaporation from the upper airways delivers an osmotic force on airway mucus that drives it toward the epithelium with increasing amount as the dryness of inhaled air increases. That part of the manuscript seems valid and supported. The connection of air dryness with climate change is suspected in general but not established. That respiratory disease and breathing abnormalities have been increasing seems a fact but attribute this solely to climate change strikes me as not supportable. Humans spend so much time indoors rather than outdoors and the air indoors is not controlled simply by climate change, so I am reluctant to recommend publication of this manuscript in its present form.

---

## [Reviewer Report]

*Comments to Author*: The authors seem unaware of the existenceof the endothelial surface layer or of the glycocalyx or of their micro/nanostructure (references enclosed) .

Essential for their problem seems to be the critical salt concentration for bubble-bubble (and nanobubble) fusion inhibition, in nanobubble stability and that gas transfer occurs viananobubbles of (N2/O2)and (CO2 /H2O) and that the structure of lung surfactants and the interaction of viruses and other pathogens are important.

A number of their issues may go away or resolved if the authors take into account the effects of these things. For example, the cystic fibrosis discussion and that about their reference 68.

Reviewer #1: I have examined with great interest the manuscript, “Breathing dry air promotes inflammation, activates neural pathways and disrupts clearance by osmotic stresses imparted on airway epithelia” because it claims to explain the rise of respiratory diseases which the authors attribute to climate change. This is attributed to chronic dehydration of the airways. To me, this is a bold claim. I do not find it supported by the evidence presented.

So, let’s turn to what is evidence is presented. It is found that water evaporation from the upper airways delivers an osmotic force on airway mucus that drives it toward the epithelium with increasing amount as the dryness of inhaled air increases. That part of the manuscript seems valid and supported. The connection of air dryness with climate change is suspected in general but not established. That respiratory disease and breathing abnormalities have been increasing seems a fact but attribute this solely to climate change strikes me as not supportable. Humans spend so much time indoors rather than outdoors and the air indoors is not controlled simply by climate change, so I am reluctant to recommend publication of this manuscript in its present form.